# Teaching Responsible Machine Learning to Engineers

**Hilde J.P. Weerts** [1]  **Mykola Pechenizkiy** [1]

## Abstract

With the increasing application of machine learning in practice, there is a growing need to incorporate ethical considerations in engineering curricula. In this paper, we reflect upon the development of a course on responsible machine learning for undergraduate engineering students. We found that technical material was relatively easy to grasp when it was directly linked to prior knowledge on machine learning. However, it was non-trivial for engineering students to make a deeper connection between real-world outcomes and ethical considerations such as fairness. Moving forward, we call upon educators to focus on the development of realistic case studies that invite students to interrogate the role of an engineer.

## 1. Introduction

As machine learning models are increasingly applied in practice, there is a growing interest in their responsible development and use. Although humanities scholars have studied the ethical implications of artificial intelligence for decades, the widespread application of machine learning techniques has opened up new avenues for studying the interaction between intelligent systems and society. At the same time, major machine learning venues have attracted manuscripts that address technical challenges of formulating and achieving fairness and explainability.

Within and across research communities there is an increased understanding that applying machine learning responsibly is a sociotechnical challenge that should be addressed from a multidisciplinary perspective (e.g., Raji et al., 2021). This sentiment is illustrated in an emergence of new cross-disciplinary conferences (most notably FAccT[1],

FORC[2], and AIES[3]) and specialized workshops (e.g., Bias and Fairness in AI (Calders et al., 2021)).

It seems imperative that practitioners understand in what ways machine learning models may pose ethical risks and how these risks can be mitigated. Indeed, there is a growing interest to incorporate responsible design in computer science education (Zegura et al., 2020; Raji et al., 2021; Fiesler et al., 2021). However, education on topics of fairness, accountability, confidentiality, and transparency (FACT) geared toward engineers is still in its infancy.

**Responsible Machine Learning Education**    In some programs, ethical considerations are covered as a stand-alone course, emphasizing normative ethical theories. In other programs, ethics may be incorporated as a seminar following a more technical module. While these classes are valuable, they are at risk of divorcing ethical considerations from technical practice (Malazita & Resetar, 2019; Fiesler et al., 2021). As a result, students may have a hard time applying ethical considerations in their daily professional practice (Fiesler et al., 2021).

Instead, we believe there is a need to teach responsible machine learning in a way that (1) encourages students to engage with ethical considerations of machine learning systems, (2) is applicable to the daily practice of engineers, i.e., provides concrete and actionable pointers. With these goals in mind, we have designed a new course, *Responsible Machine Learning* (RML), at Eindhoven University of Technology in the Netherlands, targeted primarily at final-stage undergraduate engineering students majoring in either data science or computer science. We expect the course to be equally suitable for students of other technical disciplines, such as statistics or theoretical computer science, provided they have prior experience with machine learning and programming.

In this paper, we detail the instructional design of the course and reflect upon our experiences. Although RML covered various topics, we will limit our discussion mostly to teaching algorithmic fairness. In the remainder of this paper, we assume the reader is familiar with basic concepts of algo-

---

[1]Eindhoven University of Technology, The Netherlands. Correspondence to: Hilde Weerts <h.j.p.weerts@tue.nl>.

*Proceedings of the $2^{nd}$ Teaching in Machine Learning Workshop*, PMLR, 2021. Copyright 2021 by the author(s).

[1]https://facctconference.org/

[2]https://responsiblecomputing.org/forc-2021-program/

[3]https://www.aies-conference.com/

rithmic fairness (see e.g., Chouldechova & Roth (2020) for a recent survey).

**Lessons Learned** After the first course iteration, we found that technical material was relatively easy to grasp for our target audience when it was directly linked to prior knowledge on machine learning. In particular, we found toy examples, demos, and tutorials to be useful tools to foster student understanding.

However, we have also noticed that it was non-trivial for our students to make a deeper connection between real-world outcomes and algorithmic fairness. One of the main challenges in teaching RML was to simplify a complex topic to facilitate understanding, without reducing it to a narrow, technical perspective. To this end, realistic and concrete case studies as well as invited lectures were helpful.

**Moving Forward** Despite the raising level of public and academic discourse, high-quality educational resources suitable for undergraduate engineering students are scarce. Moving forward, we call upon educators to develop more realistic and concrete case studies, allowing engineering students to connect ethical considerations and technical decision-making in a more meaningful way.

**Outline** The remainder of this paper is structured as follows. In Section 2, we describe our course design and reflect upon our experiences. In Section 3, we sketch paths for future work.

## 2. Course Design

Following the principles of constructive alignment (Kandlbinder et al., 2014), our course design consists of three components: learning objectives, learning activities, and assessment. Due to COVID-19 restrictions, the course was taught fully online.

### 2.1. Learning Objectives

RML is structured around four main themes: Fairness, Accountability, Confidentiality, and Transparency (FACT). Of these themes, fairness and transparency are covered most extensively. The learning objectives of the course were as follows.

*At the end of the course, students will be able to:*

**1.** *Evaluate and communicate trade-offs between (socio)technical desiderata of machine learning applications, taking into account diverse stakeholders' perspectives.*

**2.** *Explain technical and organizational strategies for advancing FACT throughout the machine learning development process.*

**3.** *Select and implement appropriate strategies for enhancing algorithmic fairness and interpretable/explainable machine learning.*

We would like to highlight a few aspects of these objectives. First of all, learning objective 1 emphasizes *communicating* trade-offs. Even well-intentioned practitioners can contribute to harmful technology through implicit design choices. By making trade-offs more explicit, they can be discussed with other stakeholders, fostering accountability. Second, learning objective 1 emphasizes engaging diverse *stakeholders*, the importance of which as been stressed previously by e.g., Raji et al. (2021). Third, learning objective 2 highlights how different strategies can be applied *throughout* the machine learning development process - not just as an afterthought. And finally, learning objective 3 requires students to *implement* technical evaluation and mitigation strategies, marrying ethical considerations with the daily practice of an engineer.

### 2.2. Teaching Materials

We have found that high-quality teaching materials geared towards undergraduate engineering students are scarce. Although there exist several graduate-level courses that cover FACT topics in a research seminar format, we consider this format less suitable for undergraduate students. First of all, undergraduate students may not be able to fully grasp highly technical papers. Second, critical position papers typically assume a level of familiarity with the research field that cannot be expected from undergraduate students.

For RML, we have tried to fill this gap through the development of lectures, lecture notes (Weerts, 2021), and tutorials[4]. Additionally, assigned reading included several chapters of Barocas et al. (2019) (an incomplete work in progress at the time) and Kamiran et al. (2013a).

#### 2.2.1. SYLLABUS

We start the course with an introduction to a responsible machine learning process, structured around the CRISP-DM process model (Wirth & Hipp, 2000). In accordance to learning objective 1, our introduction emphasizes the importance of the machine learning problem understanding stage. Is this the right problem to solve? Who are the stakeholders of the envisioned system? In particular, we exemplify different types of harm, structured against the moral values they go against (e.g., safety, fairness, transparency, autonomy).

The second module of the course revolves around fairness of machine learning algorithms and the challenges associated with this (learning objectives 2 and 3). We cover several fairness metrics and mitigation algorithms coined by the

---

[4]https://github.com/hildeweerts/
2IX30-Responsible-Data-Science

machine learning community and discuss their limitations. To facilitate a deeper understanding of the relationship between fairness and technical design choices, we have also developed several tutorials in the form of Jupyter notebook (Kluyver et al., 2016), revolving around a case study of Propublica's analysis of COMPAS (Angwin et al., 2016) leveraging modules of the Python library Fairlearn (Bird et al., 2020). Although these notebooks contain code, their primary purpose is to help students consider the applicability and limitations of fairness metrics and mitigation algorithms. Finally, invited lectures of both researchers and practitioners engaged students with contemporary research discussions and showcased challenges data scientists face in practice.

### 2.2.2. Fairness as An Optimization Problem

We found that connecting fairness metrics and algorithms with prior knowledge on machine learning helped students to understand technical details. In particular, the usage of toy examples, demos, and code tutorials seemed to increase student understanding.

Many technical approaches aimed at achieving fairness can be framed as an optimization problem (Zafar et al., 2019). Through this lens, the goal is to maintain good predictive performance while satisfying fairness constraints. This can be achieved via several techniques including fairness-aware representation learning (Zemel et al., 2013; Hu et al., 2020), regularization, or post-processing of specific (Kamiran et al., 2010) or any (Hardt et al., 2016) trained models or model outputs. Mastering these topics becomes easier if a student has recently learned about concepts such as cost-sensitive learning. Similarly, prior understanding of trade-offs between predictive performance metrics (e.g., ROC-curve analysis) helps to better understand other trade-offs, such as a fairness-accuracy trade-off or conflicting notions of fairness.

We experienced difficulty in teaching counterfactual fairness (Kusner et al., 2017) in a compact way, as the majority of our students have not previously studied causal inference. However, exemplifying Simpson's paradox[5] in the context of measuring and reasoning about group fairness, and introducing the notion of explainable discrimination (Kamiran et al., 2013b) helped students to understand the limitations of purely statistical approaches and the need for taking a causal perspective.

### 2.2.3. Fairness as a Sociotechnical Challenge

As we will expand upon in Section 2.3, it was non-trivial for students to connect technical design choices and real-world outcomes. As such, one of the main challenges in

[5]Simpson's paradox (Simpson, 1951) is a phenomenon in which an association between two variables in a population disappears or reverses when it is analyzed within subgroups.

developing teaching materials was to simplify a complex, sociotechnical challenge like fairness into something that can be understood by our target audience, without reducing it to a narrow, technical perspective.

For example, historical biases may be encoded in data, which can result in downstream allocation harms. While this is important to understand, reducing unfairness to "bias in, bias out" foregoes many more fundamental questions, such as whether a predictive model should exist at all. Similarly, after covering fairness metrics, an often-heard question is "which fairness metric should I use?". The answer to this question highly depends on the context of an application. To some extent, reducing the complexity of these challenges through general frameworks seems unavoidable, but risks only a surface-level student engagement with a context.

### 2.3. Assessment

The assessment of RML consisted of three components, an individual assignment (20%), three quizzes (15%), and a final group project (65%). As most of our findings relate to the individual assignment and group project, we will limit our discussion to these.

### 2.3.1. Individual Assignment

Following the first module, students practiced identifying risks and balancing trade-offs of machine learning systems (learning objective 1) in the form of an individual assignment. The assignment was inspired by Zegura et al. (2020), who developed two role playing activities in which students need to decide upon the deployment of a system. In RML, the individual assignment was in the form of an individual report covering two scenarios, complemented by two group discussions after which students could revise their report.

The group discussions served to practice communicating trade-offs and exchanging views with peers. As our target audience is generally not familiar with instructional formats involving group discussions, we provided students with a suggested timing, meeting roles, and discussion guidelines.

The majority of students appreciated the group discussion format, as it allowed them to gain new insights. This was reflected in their reports: most students were able to identify relevant stakeholders and high-level benefits and risks. However, students had more difficulty with the precise formulation of risks and mitigation strategies. For example, students would write that the system "should be fair for all patients" or "without bias against minority groups" without exemplifying what "fair" or "without bias" entailed in this specific scenario. Similarly, students sometimes had difficulty connecting RML design choices to the identified risks, reflected in ambiguous phrasing of how mitigation strategies might alleviate some of the risks.

### 2.3.2. GROUP PROJECT

For the final assessment, we have taken a problem-based learning approach (De Graaf & Kolmos, 2003). In teams of five, students went through all stages of the machine learning development process (except for deployment) and implemented techniques for enhancing fairness and explainability (learning objective 3).

The development of a suitable project was highly non-trivial. We believe that developing a realistic prototype is crucial for students to fully appreciate the challenges of responsible design from the perspective of an engineer. As such, we set out to find a suitable real-world data set accompanied by a realistic scenario. Fairness assessments involve sensitive data, which made it challenging to find an external partner willing to collaborate in the context of undergraduate course work. Additionally, bench-marking data sets that are routinely used in fairness research often lack the necessary context (e.g., the UCI Adult data set) or relate to contested applications of machine learning (e.g., Propublica's COMPAS data set, see Bao et al. (2021)).

Eventually, we settled upon the MIMIC-Extract data set (Wang et al., 2020), a partly preprocessed data set built upon the critical care database MIMIC-III (Johnson et al., 2016). The associated task was the development of an ICU mortality prediction model that could be used as decision-support tool for physicians. In the assignment, the tool was positioned as a potential alternative to the well-established Sequential Organ Failure Assessment (SOFA) scores.

By design, the assignment was relatively open-ended. Although the scenario hinted towards fairness and transparency, no explicit requirements were given. Instead, students were required to identify requirements through their analysis of the context. To emphasize the importance of the problem formulation, a large proportion of points was awarded to this part of the assignment (learning objective 1). To teach the importance of fostering accountability, students were also required to fill out a data sheet (Gebru et al., 2018) and model card (Mitchell et al., 2019). Finally, students were asked to reflect upon their findings and (ethical) implications of limitations of their developed model.

In the course evaluation, some students indicated that they struggled with the open-ended nature and independent planning of the project. For future course iterations, we plan to provide more guidance regarding the planning and report. Nevertheless, we found that many students highly appreciated the project. Most groups were able to successfully apply various machine learning techniques, including fairness assessment and techniques for enhancing explainability. However, some groups were not able to articulate the relevance of these approaches precisely in the given context. For example, students were able to successfully compute a set of fairness metrics, but did not explain convincingly why the metrics were suitable for the problem at hand.

## 3. Moving Forward

With the design of RML, we set out to build a bridge between ethical and technical perspectives, in a way that speaks to engineers. In this paper, we have showcased our approach and reflected upon our experiences. However, much work remains to be done.

### 3.1. Realistic Case Studies

Although there is an increasing number of examples that showcase how machine learning models can be harmful, it can be difficult for students to connect technical decision-making with ethical implications beyond surface-level observations. As such, we believe realistic, detailed, and concrete case studies are crucial to facilitate student learning.

However, it has proven difficult to develop these materials within the context of a single university course. Publicly available data sets often lack the required contextualization, such as a datasheet (Gebru et al., 2018) or a realistic use case. Some of these issues might be alleviated through the use of carefully crafted synthetic data. However, that would still not allow students to engage with stakeholders' perspectives in a meaningful way and instead leave them to rely on their own assumptions. One way forward would be to expand a case study not only with a description of the scenario, but also with direct input from (potentially fictional) stakeholders. For example, we could provide students with video-recorded or transcribed interviews.

### 3.2. Interrogating the Role of an Engineer

Ethical development of machine learning is a sociotechnical challenge that cannot be solved by engineers alone. In our view, engineering students should not be expected to be well-versed in all these different disciplines. Instead, we believe it is important to show students the limitations of the computer science lens and present concrete approaches to invite other perspectives.

Therefore, we call on educators to develop more examples of multidisciplinary work that showcase the role of an engineer in relation to other actors. For example, Raji et al. (2021) suggest to develop frameworks to cooperate with peers from other disciplines and to engage with affected populations. At engineering universities, organizing team work with other disciplines can be impractical. A different way to reflect the importance of other disciplines in course work would be to give students the opportunity to consult external experts, possibly in the form of auxiliary materials that are only provided on demand.

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
