# OpenReview forum: "Teaching Responsible Machine Learning to Engineers"
_ecmlpkdd.org/ECMLPKDD/2021/Workshop/TeachML — TeachML 2021_

### Official Review · Reviewer_wDuw · 2021-07-09
**A well thought out curriculum for undergraduate engineering students**

**Rating:** 8
**Confidence:** 4

**Review:**

This paper describes innovations and lessons learned from a new course on Responsible Machine Learning, designed for undergraduate engineers. The course covers Fairness, Accountability, Confidentiality and Transparency, with the main foci being Fairness and Transparency.

There is often a tendency in ML to try to "operationalize" fairness and develop a set of rules to be followed, and I appreciated that, while the authors did highlight optimization-based strategies and checks and balances on data, much of the material highlighted ethics and the messy reality that fairness cannot be reduced to optimizing an equalized opportunity penalty. The fact that the authors described that this was the hardest part for students to grasp serves to highlight its importance.

I liked that the first module covers types of harm before getting in to technical material. I assume that the first assessment---the individual report---comes at the end of this module? It seems a good order to ensure students are thinking about what consequences might be before jumping to solutions (which might not be the right solutions). I would have liked to have seen a little more detail of what case studies were used in this module---the authors note that many students still had a hazy notion of "fairness".

The second module focuses more on technical material, such as bias mitigation approaches. The authors don't mention any technical content for explainability or confidentiality, although the description of the students' projects suggests that they incorporated some technical content at least on the former. Extrapolating from this assumption, it sounds like there is a lot of material covered, and my fear would be that students become overwhelmed---did the authors find this to be the case? Do they intend to continue to include counterfactual fairness, which is highlighted as something students struggled with?

I really liked the group project set-up, and appreciated the thoughtful dataset choice and the open-ended assignment, as this mimics real-world scenarios where engineers will have to decide what is important for a given scenario.

The main thing I would liked to have seen more of is, what specific challenges are associated with the audience being engineers?  Would this course be equally suited for a statistics or CS curriculum (perhaps with adapting the technical content to complement previous courses)? Is there an inherent assumption that the students will be engaging with issues involving FACT from the position of a software engineer or machine learning engineer, rather than say a data scientist, manager, UX researcher etc, or is the use of the term engineer simply to describe their educational background up to this point? It appears that the main challenge faced was the lack of experience and familiarity with discussing and analysing societal and ethical matters and a lack of, for want of a better word, "soft skills" such as participating in group discussions---something that would also impact other technical undergraduate courses.

---

### Official Review · Reviewer_81jo · 2021-07-09
**review of: Teaching responsible ML to engineers**

**Rating:** 8
**Confidence:** 5

**Review:**

This was an interesting and well written paper covering the very important topic of how to teach responsible machine learning to engineering students. I think this is highly relevant to the conference and should be included.

I have just a few minor suggestions for improvements:

* Do you have any quantitative metrics to show how well received the course was by the students?
* In section 2.2.2 (FAIRNESS AS AN OPTIMIZATION PROBLEM) Simpson's Paradox is mentioned but not defined or referenced. Please include a definition and/or reference to this as some readers might not be familiar with it.
* Was there anything students did not like about the course?
* I realise that not enough time may have passed yet to answer this, but I would be really interested to know how this course is perceived by employers. Does it make graduates more questioning of the orders they are given? Has it lead to changes in the approach taken by any companies employing students who took this course?

---

### Decision · Program_Chairs · 2021-07-21

**Decision:**

Accept

**Comment:**

Congratulations! The reviewers agree that this paper should be accepted.

Camera-ready version is due August 18, 2021. As you prepare the camera ready version, please take the reviewers comments into consideration.

We look forward to your participation at the workshop on September 13, 2021. We invite you also to join us for the satellite event on September 08, 2021. Schedules for both the workshop and the satellite event will be forthcoming.